# Private Online Learning via Lazy Algorithms

**Hilal Asi**
Apple
hilal.asi94@gmail.com

**Tomer Koren**
Tel Aviv University
tkoren@tauex.tau.ac.il

**Daogao Liu**[*]
University of Washington
liudaogao@gmail.com

**Kunal Talwar**
Apple
kunal@kunaltalwar.org

## Abstract

We study the problem of private online learning, focusing on online prediction from experts (OPE) and online convex optimization (OCO). We propose a new transformation that translates lazy, low-switching online learning algorithms into private algorithms. We apply our transformation to differentially private OPE and OCO using existing lazy algorithms for these problems. The resulting algorithms attain regret bounds that significantly improve over prior art in the high privacy regime, where $\varepsilon \ll 1$, obtaining $O(\sqrt{T \log d} + T^{1/3} \log(d)/\varepsilon^{2/3})$ regret for DP-OPE and $O(\sqrt{T} + T^{1/3}\sqrt{d}/\varepsilon^{2/3})$ regret for DP-OCO. We complement our results with a lower bound for DP-OPE, showing that these rates are optimal for a natural family of low-switching private algorithms.

## 1   Introduction

Online learning is a fundamental problem in machine learning, where an algorithm interacts with an oblivious adversary for $T$ rounds. First, the oblivious adversary chooses $T$ loss functions $\ell_1, \ldots, \ell_T : \mathcal{X} \to \mathbb{R}$ over a fixed decision set $\mathcal{X}$. Then, at any round $t$, the algorithm chooses a model $x_t \in \mathcal{X}$, and the adversary reveals the loss function $\ell_t$. The algorithm suffers loss $\ell_t(x_t)$, and its goal is to minimize its cumulative loss compared to the best model in hindsight, namely its *regret*:

$$\mathbf{Reg}_T = \sum_{t=1}^{T} \ell_t(x_t) - \min_{x^\star \in \mathcal{X}} \sum_{t=1}^{T} \ell_t(x^\star).$$

In this work, we study two different *differentially private* instances of this problem: differentially private online prediction from experts (DP-OPE) where the model $x$ can be chosen from $d$ experts ($\mathcal{X} = [d]$); and differentially private online convex optimization (DP-OCO) where the model belongs to a convex set $\mathcal{X} \subset \mathbb{R}^d$.

Both problems have been extensively studied recently [JKT12, ST13, JT14, AS17, KMS$^+$21] and an exciting new direction with promising results for this problem is that of designing private algorithms based on low-switching algorithms for online learning [AFKT23b, AFKT23a, AKST23a, AKST23b]. The main idea in these works is that the privacy cost for privatizing a low-switching algorithm can be significantly smaller as these algorithms do not update their models too frequently, allowing them to allocate a larger privacy budget for each update. This has been initiated by [AFKT23b], which used the shrinking dartboard algorithm to design new algorithms for DP-OPE, later revisited by [AKST23a] to design new algorithms for DP-OCO using a regularized follow-the-perturbed-leader approach, and more recently by [AKST23b] which used a lazy and regularized version of the multiplicative weights algorithm to obtain improved rates for DP-OCO.

---

[*]Part of this work was done while interning at Apple.

38th Conference on Neural Information Processing Systems (NeurIPS 2024).

| | Prior work | This work |
|---|---|---|
| **DP-OPE** | $\sqrt{T\log d} + \frac{\min\{\sqrt{d}, T^{1/3}\}\log d}{\varepsilon}$ [AS17, AFKT23b] | $\sqrt{T\log d} + \frac{T^{1/3}\log d}{\varepsilon^{2/3}}$ |
| **DP-OCO** | $\min\left\{\frac{d^{1/4}\sqrt{T}}{\sqrt{\varepsilon}}, \sqrt{T} + \frac{T^{1/3}\sqrt{d}}{\varepsilon} + \frac{T^{3/8}\sqrt{d}}{\varepsilon^{3/4}}\right\}$ [KMS$^+$21, AKST23b] | $\sqrt{T} + \frac{T^{1/3}\sqrt{d}}{\varepsilon^{2/3}}$ |

Table 1: Regret for approximate $(\varepsilon, \delta)$-DP algorithms. For readability, we omit logarithmic factors that depend on $T$ and $1/\delta$.

While all of these results build on lazy-switching algorithms for designing private online algorithms, each one of them has a different method for achieving privacy and, to a greater extent, a different analysis. Moreover, it is not clear whether these transformations from lazy to private algorithms in prior work have fulfilled the full potential of lazy algorithms for private online learning and whether better algorithms are possible through this approach. Indeed, the regret obtained in prior work [AFKT23b, AKST23b] is $T^{1/3}/\varepsilon$ (omitting dependence on $d$) for DP-OPE, which implies that the normalized regret is $1/T^{2/3}\varepsilon$: this is different than what exhibited in a majority of scenarios of private optimization, where the normalized error is usually a function of $T\varepsilon$.

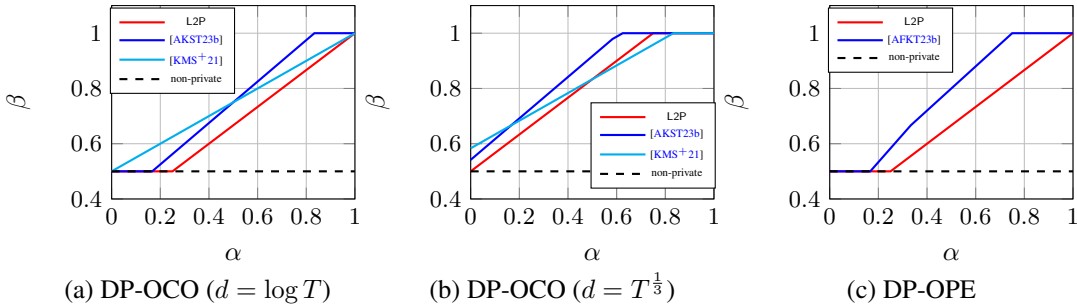

(a) DP-OCO ($d = \log T$)  (b) DP-OCO ($d = T^{\frac{1}{3}}$)  (c) DP-OPE

Figure 1: Regret bounds for (a) DP-OCO with $d = \text{poly}\log(T)$, (b) DP-OCO with $d = T^{1/3}$ and (c) DP-OPE with $d = T$. We denote the privacy parameter $\varepsilon = T^{-\alpha}$ and regret $T^{\beta}$, and plot $\beta$ as a function of $\alpha$ (ignoring logarithmic factors).

## 1.1 Our contributions

Our main contribution in this work is a new transformation that converts lazy online learning algorithms into private algorithms with similar regret guarantees, resulting in new state-of-the-art rates for DP-OPE and DP-OCO. We provide a summary in Table 1 and Figure 1.

**L2P: a transformation from lazy to private algorithms (Section 3).** Our main contribution is a new transformation, we call L2P, that allows converting any lazy algorithm into a private one with only a slight cost in regret. This allows us to use a long line of work on lazy online learning [KV05, GVW10, AT18, CYLK20, SK21, AKST23a] to design new algorithms for the private setting. Our transformation builds on two new techniques: first, we design a new switching rule that only depends on the loss at the current round, so as to minimize the privacy cost of each switching and mitigate the accumulation of privacy loss. Second, we rely on a simple, key observation that by grouping losses in a large batch, we can minimize the effect on the regret of lazy online learning algorithms. We introduce a new analysis for the regret of lazy online algorithms with a large batch size that improves over the existing analysis in [AFKT23b]; this allows us to reduce the total number of "fake switches" needed to guarantees privacy, improving the final regret.

**Faster rates for DP-OPE (Section 3.1).** As a first application, we use our transformation in the DP-OPE problem on the multiplicative weights algorithm [AHK12]. This results in a new algorithm for DP-OPE that has regret $\sqrt{T\log(d)} + T^{1/3}\log(d)/\varepsilon^{2/3}$, improving over the best existing results for the high-dimensional regime in which the regret is $\sqrt{T\log(d)} + T^{1/3}\log(d)/\varepsilon$ [AFKT23b].[2]

---

[2][AFKT23b] has another algorithm which slightly improves over this regret in the high-privacy regime and obtains regret $T^{2/5}/\varepsilon^{4/5}$. We include this algorithm in Figure 1.

The improvement is particularly crucial in the high-privacy regime, where $\varepsilon \ll 1$: indeed, our regret shows that (for $d = \mathsf{poly}(T)$) it is sufficient to set $\varepsilon \geq T^{-1/4}$ for matching the optimal non-private regret $\sqrt{T \log d}$, whereas previous results require a much larger $\varepsilon \geq T^{-1/6}$ to get privacy for free. This is also important in practice, when multiple applications of DP-OPE are necessary: using advanced composition, our result shows that we can solve $K \approx \sqrt{T}$ instances of DP-OPE with $\varepsilon = 1$ and still obtain the non-private regret of order $\sqrt{T}$; in contrast, prior work only allows to solve $K \approx T^{1/3}$ instances while still attaining the non-private regret.

**Faster rates for DP-OCO (Section 3.2).** As another application, we use our transformation for DP-OCO with the regularized multiplicative weights algorithm of [AKST23b]. We obtain a new algorithm for DP-OCO that has regret $\sqrt{T} + T^{1/3}\sqrt{d}/\varepsilon^{2/3}$, improving over the best existing results that established regret $\sqrt{T} + T^{1/3}\sqrt{d}/\varepsilon + T^{3/8}\sqrt{d}/\varepsilon^{3/4}$ [AKST23b] or $d^{1/4}\sqrt{T}/\sqrt{\varepsilon}$ using DP-FTRL [KMS+21].

**Lower bounds for low-switching private algorithms (Section 4).** To understand the limitations of low-switching private algorithms, we prove a lower bound for the natural family of private algorithms with limited switching, showing that the upper bounds obtained via our reduction are nearly tight for this family of algorithms up to logarithmic factors. This shows that new techniques, beyond limited switching, are required in order to improve upon our upper bounds.

**Related work.** Our transformation and algorithms build on a long line of work in online learning with limited switching [KV05, GVW10, AT18, CYLK20, SK21, AKST23b]. As is evident from prior work in private online learning, the problems of lazy online learning and private online learning are tightly connected [AFKT23b, AFKT23a, AKST23a, AKST23b]. In this problem, the algorithm wishes to minimize its regret while making at most $S$ switches: the algorithm can update the model at most $S$ times throughout the $T$ rounds. Recent work has resolved the lazy OPE problem: [AT18] show a lower bound of $\sqrt{T} + (T/S)\log(d)$ on the regret, which is achieved by several algorithms such as Follow-the-perturbed-leader [KV05] and the shrinking dartboard algorithm [GVW10]. For lazy OCO, however, optimal rates are yet to be known: [AKST23b] recently show that a lazy version of the regularized multiplicative weights algorithm obtains regret $\sqrt{T} + (T/S)\sqrt{d}$, whereas the best lower bound is $\sqrt{T} + T/S$ [SK21].

## 2 Preliminaries

### 2.1 Problem setup

We consider an interactive $T$-round game between an algorithm ALG and an oblivious adversary Adv. Before the interaction, the adversary Adv chooses $T$ loss functions $\ell_1, \ldots, \ell_T \in \mathcal{L} = \{\ell \mid \ell : \mathcal{X} \to \mathbb{R}\}$. Then, at each round $t \in [T]$, the algorithm ALG, which observed $\ell_1, \cdots, \ell_{t-1}$ chooses $x_t \in \mathcal{X}$, and then the loss function $\ell_t$ chosen by Adv is revealed. The regret of the algorithm ALG is defined below:

$$\mathbf{Reg}_T(\mathsf{ALG}) := \sum_{t=1}^{T} \ell_t(x_t) - \min_{x^* \in \mathcal{X}} \sum_{t=1}^{T} \ell_t(x^*).$$

We study online optimization under the constraint that the algorithm is differentially private. For an algorithm ALG and a sequence $\mathcal{S} = (\ell_1, \ldots, \ell_T)$ chosen by an oblivious adversary Adv, we let $\mathsf{ALG}(\mathcal{S}) := (x_1, \ldots, x_T)$ denote the output of ALG over the loss sequence $\mathcal{S}$. We have the following definition of privacy against oblivious adversaries.[3]

**Definition 2.1** (Differential privacy). A randomized algorithm ALG is $(\varepsilon, \delta)$-differentially private against oblivious adversaries $((\varepsilon, \delta)$-DP) if, for all neighboring sequences $\mathcal{S} = (\ell_1, \ldots, \ell_T) \in \mathcal{L}^T$

---

[3]Our regret bound may be invalid with an adaptive adversary, but our algorithms will satisfy a stronger notion of differential privacy against adaptive adversaries (see [AFKT23b]). However, to keep the notation and analysis simpler, we limit our attention to privacy against oblivious adversaries.

and $\mathcal{S}' = (\ell'_1, \ldots, \ell'_T) \in \mathcal{L}^T$ that differ in a single element, and for all events $\mathcal{O}$ in the output space of ALG, we have

$$\Pr[\mathsf{ALG}(\mathcal{S}) \in \mathcal{O}] \leq e^\varepsilon \Pr[\mathsf{ALG}(\mathcal{S}') \in \mathcal{O}] + \delta.$$

We focus on two important instances of differentially private online optimization:

(i) **DP Online Convex Optimization (DP-OCO).** In this problem, the adversary picks loss functions $\ell \in \mathcal{L}_{OCO} := \{\ell \mid \ell : \mathcal{X} \to \mathbb{R} \text{ is convex and } L\text{-Lipschitz}\}$ where $\mathcal{X} \subset \mathbb{R}^d$ is a convex set with diameter $D = \mathsf{diam}(\mathcal{X}) := \sup_{x,y \in \mathcal{X}} \|x - y\|$, and the algorithm chooses $x_t \in \mathcal{X}$. The goal of the algorithm is to minimize regret while being $(\varepsilon, \delta)$-differentially private.

(ii) **DP Online Prediction from Experts (DP-OPE).** In this problem, the adversary picks loss functions $\ell \in \mathcal{L}_{OPE} = \{\ell \mid \ell : [d] \to [0, 1]\}$ where $\mathcal{X} = [d]$ is the set of $d$ experts, and the algorithm chooses $x_t \in [d]$. The goal of the algorithm is to minimize regret while being $(\varepsilon, \delta)$-differentially private.

## 2.2 Tools from differential privacy

Our analysis crucially relies on the following divergence between two distributions.

**Definition 2.2** ($\delta$-Approximate Max Divergence). For two distributions $\mu$ and $\nu$, we define

$$D_\infty^\delta(\mu \| \nu) := \sup_{S \subseteq \mathrm{supp}(\mu), \mu(S) \geq \delta} \ln \frac{\mu(S) - \delta}{\nu(S)}.$$

We let $D_\infty^\delta(\mu, \nu) := \max\{D_\infty^\delta(\mu \| \nu), D_\infty^\delta(\nu \| \mu)\}$.

We also use the notion of indistinguishability between two distributions.

**Definition 2.3.** $((\varepsilon, \delta)$-indistinguishability) Two distributions $\mu, \nu$ are $(\varepsilon, \delta)$-indistinguishable, denoted $\mu \approx_{(\varepsilon, \delta)} \nu$, if $D_\infty^\delta(\mu, \nu) \leq \varepsilon$.

Note that if an algorithm ALG has $\mathsf{ALG}(\mathcal{S}) \approx_{(\varepsilon, \delta)} \mathsf{ALG}(\mathcal{S}')$ for all neighboring datasets $\mathcal{S}, \mathcal{S}'$ then ALG is $(\varepsilon, \delta)$-differentially private. We direct readers to Appendix A for additional background information and detailed preliminaries.

## 3 L2P: From Lazy to Private Algorithms for Online Learning

This section presents our L2P transformation, which turns lazy online learning algorithms into private ones. The transformation has an input algorithm $\mathcal{A}$ with measure $\mu_t$ at round $t$ and samples $x_t$ from the normalized measure $\overline{\mu}_t$, which satisfies the following condition:

**Assumption 3.1.** The online algorithm $\mathcal{A}$ has at time $t$ a measure $\mu_t$ that is a function of $\ell_1, \ldots, \ell_{t-1}$ (and density function $\overline{\mu}_t$) such that for some $\delta_0 \leq 1$ and $0 < \eta \leq 1/10$ that are data-independent, we have

- $D_\infty^{\delta_0}(\overline{\mu}_{t+1}, \overline{\mu}_t) \leq \eta$,

- $\mu_{t+1}(x)/\mu_t(x) = \mathsf{func}(\ell_t, x)$ for all $x \in \mathcal{X}$ where $\mathsf{func}$ is a data-independent function.

While algorithms satisfying Assumption 3.1 need not be lazy, this assumption is satisfied by most existing lazy online learning algorithms such as the shrinking dartboard (Section 3.1) and lazy regularized multiplicative weights (Section 3.2). Moreover, any algorithm that satisfies this assumption can be made lazy via our reduction.

**Technique Overview:** Suppose the neighboring datasets differ from the $s_0$-th loss function. The high-level intuition behind our framework is that our algorithm only loses the privacy budget when it makes a switch (draws a fresh sample) whenever $t > s_0$. Hence, in the framework, we try to make the algorithm make as few switches as possible. This modification can lead to additional regret compared to lazy online learning algorithms, and we need to balance the privacy-regret trade-off. The family of

low-switching algorithms is ideal for privatization because its built-in low-switching property can achieve a better trade-off.

Our starting point is the ideas in [AFKT23b, AKST23b] to privatize low-switching algorithms, which use correlated sampling to argue that a sample from $x_{t-1} \sim \overline{\mu}_{t-1}$ is likely a good sample from $\overline{\mu}_t$ and therefore switching at round $t$ is often not necessary. In particular, at round $t$, these algorithms sample a Bernoulli random variable $S_t \sim \text{Ber}(c \cdot \overline{\mu}_t(x_{t-1})/\overline{\mu}_{t-1}(x_{t-1}))$ for some constant $c$ and use the same model $x_t = x_{t-1}$ if $S_t = 1$, and otherwise sample new model $x_t \sim \overline{\mu}_t$ if $S_t = 0$ (which happens with small probability). This guarantees that the marginal probability of the lazy iterates remains the same as the original iterates. Finally, to preserve the privacy of the switching decisions, existing algorithms add a fake switching probability $p$ where the algorithm switches independently of the input. To summarize, *existing* low-switching private algorithms work roughly as follows:

$$
\begin{cases}
\text{At each round } t: \\
\quad - \text{ Sample } S_t \sim \text{Ber}(C \cdot \overline{\mu}_t(x_{t-1})/\overline{\mu}_{t-1}(x_{t-1})) \text{ and } S'_t \sim \text{Ber}(1-p) \\
\quad - \text{ Sample new } x_t \sim \overline{\mu}_t \text{ if } S_t = 0 \text{ or } S'_t = 0 \\
\quad - \text{ Otherwise set } x_t = x_{t-1}
\end{cases}
$$

This sketch is the starting point of our transformation, and we will introduce two new components to improve performance. The first component aims to avoid the accumulation of privacy cost for switching in the current approaches where each user can affect the switching probability for all subsequent rounds: this happens since $\overline{\mu}_t(x_{t-1})/\overline{\mu}_{t-1}(x_{t-1})$ is usually a function of the whole history $\ell_1, \ldots, \ell_t$, and hence the existing low-switching private algorithms lose the privacy budget even it does not make real switches. To address this, we deploy a new correlated sampling strategy in L2P where the loss $\ell_t$ at time $t$ affects the switching probability only at time $t$, hence paying a privacy cost for switching only in a single round. To this end, we construct a parallel sequence of models $\{y_t\}_{t \in [T]}$ (independent of $x_t$) that is used for normalizing the ratio $\overline{\mu}_t(x_{t-1})/\overline{\mu}_{t-1}(x_{t-1})$ to become independent of the history. In particular, at round $t$, we switch with probability proportional to

$$
\frac{\overline{\mu}_t(x_{t-1})}{\overline{\mu}_{t-1}(x_{t-1})} \cdot \frac{\overline{\mu}_{t-1}(y_{t-1})}{\overline{\mu}_t(y_{t-1})}.
$$

The main observation here is that $\frac{\overline{\mu}_t(x_{t-1})}{\overline{\mu}_{t-1}(x_{t-1})} \cdot \frac{\overline{\mu}_{t-1}(y_{t-1})}{\overline{\mu}_t(y_{t-1})} = \frac{\mu_t(x_{t-1})}{\mu_{t-1}(x_{t-1})} \cdot \frac{\mu_{t-1}(y_{t-1})}{\mu_t(y_{t-1})}$ and this ratio is a function of $\ell_t$ our input online learning algorithms which satisfies Assumption 3.1. This will, therefore, improve the privacy guarantee of the final algorithm.

The second main observation in L2P is that having a large batch size (batching rounds together) does not significantly affect the regret of lazy online algorithms compared to non-lazy algorithms but can further reduce the times to make switches and save the privacy budget. Our main novelty is a new analysis of the effect of batching on the regret of lazy algorithms (Proposition 3.3), which states that running a lazy online algorithm with a batch size of $B$ would have an additive error of $TB^2\eta^2$ to the regret where $\eta$ is a measure of distance between $\overline{\mu}_t$ and $\overline{\mu}_{t-1}$. This significantly improves over existing analysis by [AFKT23b, Theorem 2] which shows that batching can add an additive term of $B/\eta$ to the regret.

Having reviewed our main techniques, we proceed to present the full details of our L2P transformation in Algorithm 1, denoting $\nu_s = \mu_{(s-1)B+1}$ where $B$ is the batch size.

The regret of our transformation depends on the regret of its input algorithm. For the measure $\{\mu_t\}_{t=1}^T$, we denote its regret

$$
\mathbf{Reg}_T(\{\mu_t\}_{t=1}^T) := \sum_{t=1}^T \mathbb{E}_{x_t \sim \overline{\mu}_t}[\ell_t(x_t)] - \min_{x \in \mathcal{X}} \sum_{t=1}^T \ell_t(x).
$$

The following theorem summarizes the main guarantees of Algorithm 1.

**Theorem 3.2.** *Let $p \in (0, 1)$ and $B \in \mathbb{N}$. Assuming Assumption 3.1, $Tp/B \geq 1$, and for any $\delta_1 > 0$ such that $\eta B \log(1/\delta_1)/p \leq 1$, our transformation L2P is $(\varepsilon, \delta)$-DP with*

$$
\varepsilon = \frac{2\eta}{p} + \eta + \frac{3T\eta^2 p \log(1/\delta_1)}{2B} + \sqrt{6T\eta^2 p \log^2(1/\delta_1)/B},
$$

---
**Algorithm 1:** L2P
---
1 **Input:** Parameter $\eta$, measures $\{\nu_t\}_{t\in[T]}$, batch size $B$, fake switching parameter $p$ ;
2 Sample $x_1, y_1 \sim \overline{\nu}_1$;
3 Observe $\ell_1, \ldots, \ell_B$ and suffer loss $\sum_{i=1}^{B} \ell_i(x_1)$;
4 **for** $s = 2, \cdots, T/B$ **do**
5 $\quad$ Sample $S_s \sim \mathsf{Ber}\left(\min\left(1, \frac{\nu_s(x_{s-1})}{e^{2B\eta}\nu_{s-1}(x_{s-1})} \cdot \frac{\nu_{s-1}(y_{s-1})}{\nu_s(y_{s-1})}\right)\right)$ and $S'_s \sim \mathsf{Ber}(1-p)$;
6 $\quad$ **if** $S_s = 0$ *or* $S'_s = 0$ **then**
7 $\quad\quad$ Sample $x_s \sim \overline{\nu}_s$ ;
8 $\quad$ **else**
9 $\quad\quad$ Set $x_s = x_{s-1}$;
10 $\quad$ Sample $A_s \sim \mathsf{Ber}(1-p)$;
11 $\quad$ **if** $A_s = 0$ **then**
12 $\quad\quad$ Sample $y_s \sim \overline{\nu}_s$ ;
13 $\quad$ **else**
14 $\quad\quad$ Set $y_s = y_{s-1}$;
15 $\quad$ Play $x_s$;
16 $\quad$ Observe $\ell_{(s-1)B+1}, \ldots, \ell_{sB}$ and suffer loss $\sum_{i=(s-1)B+1}^{sB} \ell_i(x_s)$;

$$\delta = 2T(2/\eta + \log(1/\delta_1)/p)eB\delta_0 + 2T\delta_1,$$

*and has regret*

$$\mathbf{Reg}_T \le \mathbf{Reg}_T(\{\mu_t\}_{t=1}^T) + O\left(TB^2\eta^2 + \frac{\delta_0 T^2 \log(\frac{1}{\delta_1})}{\eta}\right).$$

We begin by proving the utility guarantees of our transformation. It will follow directly from the following proposition, which bounds the regret of running L2P over a lazy online learning algorithm.

**Proposition 3.3** (Regret of Batched Lazy Algorithm). *Let* ALG *be an online learning algorithm that satisfies Assumption 3.1. Let* $\eta B \log(1/\delta_1)/p \le 1$, *and* $\delta_1, \eta < 1/2$. *Then running* L2P *with the input algorithm* ALG *has regret*

$$\mathbf{Reg}_T \le \mathbf{Reg}_T(\{\mu_t\}_{t=1}^T) + O\left(TB^2\eta^2 + \frac{\delta_0 T^2 \log(\frac{1}{\delta_1})}{\eta}\right).$$

To prove Proposition 3.3, we first show that we can instead analyze the utility of a simpler algorithm that samples from $\overline{\nu}_s$ at each round. This is due to the following lemma, which shows that $\|\widehat{\nu}_s - \overline{\nu}_s\|_{TV}$ is small where $\widehat{\nu}_s$ is the marginal distribution of $x_s$ in Algorithm 1.

**Lemma 3.4.** *Let* $\widehat{\nu}_s$ *be the marginal distribution of* $x_s$ *in Algorithm 1. When* $\eta B \log(1/\delta_1)/p \le 1$, *we have* $\|\widehat{\nu}_s - \overline{\nu}_s\|_{TV} \le 3(s-1)(2e + \log(1/\delta_1)/p)B\delta_0$.

We also require the following lemma which allows to build a coupling over multiple variables, such that the variables are as close as possible. This will be used to construct a coupling between the lazy algorithm and the L2P algorithm that runs it.

**Lemma 3.5** ([AS19]). *Given a collection $S$ of random variables, all absolutely continuous w.r.t. a common $\sigma$-finite measure. Then, there exists a coupling $\Gamma$, such that for any variables $X, Y \in S$, we have* $\Pr[X \ne Y] \le \frac{2\|X-Y\|_{TV}}{1+\|X-Y\|_{TV}}$.

We are now ready to prove Proposition 3.3

*Proof.* Let $\mathbf{Reg}'_T$ denote the regret when the marginal distribution of $x_t$ is $\overline{\nu}_t$ instead of $\widehat{\nu}_t$ induced in the Algorithm. Since each loss function is bounded,

$$\mathbf{Reg}_T \le \mathbf{Reg}'_T + B \sum_{s\in[T/B]} \|\overline{\nu}_s - \widehat{\nu}_s\|_{TV}.$$

By Lemma 3.4, we have

$$\mathbf{Reg}_T \le \mathbf{Reg}'_T + B \sum_{s \in [T/B]} 3(s-1)(2/\eta + \log(1/\delta_1)/p) eB\delta_0$$

$$\le \mathbf{Reg}'_T + 8T^2 \delta_0 \log(1/\delta_1)/\eta.$$

Thus, it now suffices to upper bound $\mathbf{Reg}'_T$.

Due to the preconditions that $D_\infty^{\delta_0}(\overline{\mu}_{i+1}, \overline{\mu}_i) \le \eta$ and $\delta_0 \le \eta$, we know $\|\overline{\mu}_{i+1} - \overline{\mu}_i\|_{TV} \le 2\eta$. Recall that we assume $x_s \sim \overline{\nu}_s$. Suppose $z_i$ is the action taken by the input lazy algorithm $\mathcal{A}$ for $i \in [T]$ and the marginal distribution of $z_i$ is $\overline{\mu}_i$. By Lemma 3.5, we can construct a coupling $\Gamma_s$ between $x_s$ and $\overline{z} := (z_{(s-1)B+1}, \cdots, z_{sB})$, such that

$$\Pr_{(x_s, \overline{z}) \sim \Gamma_s}[\exists i \in [(s-1)B+1, sB], z_i \ne x_s] \le B\eta.$$

Letting $I_s = \mathbf{1}(\exists i \in [(s-1)B+1, sB], z_i \ne x_s)$, we have

$$\mathbb{E}_{x_s \sim \overline{\nu}_s} \sum_{i=(s-1)B+1}^{sB} \ell_i(x_s) = \mathbb{E}_{(x_s, \overline{z}) \sim \Gamma_s} \sum_{i=(s-1)B+1}^{sB} \ell_i(x_s)$$

$$= \mathbb{E}_{x_s, \overline{z} \sim \Gamma_s} (1 - I_s) \sum_{i=(s-1)B+1}^{sB} \ell_i(z_i)$$

$$+ \mathbb{E}_{x_s, \overline{z} \sim \Gamma_s} I_s \sum_{i=(s-1)B+1}^{sB} \ell_i(x_s)$$

$$\le \mathbb{E}_{x_s, \overline{z} \sim \Gamma_s} (1 - I_s) \sum_{i=(s-1)B+1}^{sB} \ell_i(z_i)$$

$$+ \mathbb{E}_{x_s, \overline{z} \sim \Gamma_s} I_s \sum_{i=(s-1)B+1}^{sB} (\ell_i(z_i) + O(B\eta))$$

$$\le \mathbb{E}_{z_i \sim \overline{\mu}_i} \sum_{i=(s-1)B+1}^{sB} \ell_i(z_i) + O(B\eta \cdot B^2\eta).$$

Hence we get $\mathbf{Reg}'_T \le \mathbf{Reg}_T(\{\mu_t\}_{t=1}^T) + \frac{T}{B} \cdot O(B^3\eta^2)$, which completes the proof. $\square$

Now we turn to prove the privacy of L2P. We begin with the following lemma, which provides the privacy guarantees of sampling a new model $x_t$ from the distribution $\mu_t$. We defer the proof to Appendix B.

**Lemma 3.6.** *Let $\{\mu_t\}_{t=0}^T$ satisfy Assumption 3.1 where $\eta \le 1/10$. Then for any neighboring sequences $\mathcal{S}$ and $\mathcal{S}'$ with corresponding $\{\mu_t\}_{t=0}^T$ and $\{\mu'_t\}_{t=0}^T$ that differ one loss function, we have*

$$D_\infty^{4\delta_0}(\overline{\mu}_t, \overline{\mu}'_t) \le 2\eta.$$

We use correlated sampling in the algorithm rather than sampling from $x_t$ directly. To this end, we need the following lemma, which provides upper and lower bounds on the ratio used for correlated sampling.

**Lemma 3.7.** *For any $s \in [T/B]$, if $\eta B \log(1/\delta_1)/p \le 1$, then with probability at least $1 - (2/\eta + \log(1/\delta_1)/p) \cdot eB\delta_0 - \delta_1$,*

$$\frac{\nu_{s+1}(x_s)}{\nu_s(x_s)} \cdot \frac{\nu_s(y_s)}{\nu_{s+1}(y_s)} \in [e^{-2B\eta}, e^{2B\eta}].$$

The privacy proof will build on the previous two lemmas to control the privacy cost of updating the model and the cost of the switching time. We defer the proof to Appendix B.

One remaining issue is we need to conditional on the high probability events in Lemma 3.7 for the privacy guarantee and can not directly apply Advanced Composition (Lemma A.3). Now, we modify the Advanced Composition for our usage. In the classic $k$-fold adaptive composition experiment, the adversary, after getting the first $i - 1$ answers $Y_1, \cdots, Y_{i-1}$ (denoted by $Y_{[i-1]}$ for simplicity), can output two datasets $D_i^0$ and $D_i^1$, a query $q_i$, and receives the answer $Y_i \sim \mathcal{M}_i(D_i^b, q_i)$ for the secret bit $b \in \{0, 1\}$. If each $\mathcal{M}_i$ is $(\varepsilon_i, \delta_i)$-DP, then the joint distributions over the answers $Y_{[k]}$ satisfy the advanced composition theorem.

In our case, however, we know there exists a subset $G_{i-1}(D_{[i-1]}^b)$, such that with probability at least $1 - \lambda_i$, $Y_{[i-1]} \in G_{i-1}(D_{[i-1]}^b)$. Conditional on $Y_{[i-1]} \in \cap_{b \in \{0,1\}} G_{i-1}(D_{[i-1]}^b)$,

$$\mathcal{M}_i(D_i^0, q_i \mid Y_{[i-1]} \in \cap_{b \in \{0,1\}} G_{i-1}(D_{[i-1]}^b)) \approx_{(\varepsilon_i, \delta_i)} \mathcal{M}_i(D_i^1, q_i \mid Y_{[i-1]} \in \cap_{b \in \{0,1\}} G_{i-1}(D_{[i-1]}^b)) \tag{1}$$

Then we have the following lemma:

**Lemma 3.8.** *Given the $k$ mechanisms satisfying the Condition* (1)*, then the class of mechanisms satisfy $(\tilde{\varepsilon}_{\tilde{\delta}}, 1 - (1 - \tilde{\delta})\Pi_{t \in [k]}(1 - \tilde{\delta}_t)) + 2\sum_{t \in [k]} \lambda_t$-DP under $k$-fold adaptive composition, with $\tilde{\varepsilon}_{\tilde{\delta}}$ defined in Equation* (4)*.*

*Proof.* Without losing generality, suppose we know the adversary and how they generate the databases and queries. We can construct a series of mechanisms $\mathcal{M}_i'$, such that $\mathcal{M}_i'$ draws $Y_i$ from $\mathcal{M}_i(D_i^b, q_i)$, and outputs $Y_i$ if $Y_i \in \cap_{b \in \{0,1\}} G_{i-1}(D_{[i-1]}^b)$, and outputs $\mathbf{0}$ otherwise. Let $(Y_{1,b}', \cdots, Y_{k,b}')$ be the outputs of $\mathcal{M}_i'$ with secret bit $b$, and we know the TV distance between $(Y_{1,b}', \cdots, Y_{k,b}')$ and $(Y_{1,b}, \cdots, Y_{k,b})$ is at most $\sum_{t \in [k]} \lambda_t$ for any $b \in \{0, 1\}$. Moreover, we know

$$(Y_{1,0}', \cdots, Y_{k,0}') \approx_{\tilde{\varepsilon}_{\tilde{\delta}}, 1 - (1 - \tilde{\delta})\Pi_{t \in [k]}(1 - \tilde{\delta}_t))} (Y_{1,1}', \cdots, Y_{k,1}')$$

by the advanced composition. The basic composition finishes the proof. $\qquad\square$

## 3.1 Application to DP-OPE

This section discusses the first application of our transformation to differentially private online prediction from experts (DP-OPE). Towards this end, we apply our transformation over the multiplicative weights algorithms [AHK12], which can be made lazy as done in the shrinking dartboard algorithm [GVW10]. It has the following measure at round $t$

$$\mu_t^{\mathsf{mw}}(x) = e^{-\eta \sum_{i=1}^{t-1} \ell_i(x)}. \tag{2}$$

The following proposition shows that this measure satisfies the desired properties required by our transformation. We let $\overline{\mu}_t^{\mathsf{mw}}$ denote the density corresponding to $\mu_t^{\mathsf{mw}}$.

**Lemma 3.9.** *Assume $\ell_1, \ldots, \ell_T$ where $\ell_t : [d] \to [0, 1]$. Then we have that*

1. $D_\infty^{\delta_0}(\overline{\mu}_{t+1}^{\mathsf{mw}}, \overline{\mu}_t^{\mathsf{mw}}) \leq \eta$ *with $\delta_0 = 0$.*

2. $\frac{\mu_{t+1}^{\mathsf{mw}}(x)}{\mu_t^{\mathsf{mw}}(x)} = e^{-\eta \ell_t(x)}$ *for all $x \in [d]$.*

*Proof.* The first item follows from the guarantees of the exponential mechanism as $\ell_t(x) \in [0, 1]$ for all $x \in [d]$. The second item follows immediately from the definition of $\mu^{\mathsf{mw}}$. $\qquad\square$

Having proved our desired properties, our transformation now gives the following theorem.

**Theorem 3.10** (DP-OPE). *Let $\ell_1, \ldots, \ell_T$ where $\ell_t : [d] \to [0, 1]$. Setting $B = 1/\varepsilon$ and $\eta = \min(\varepsilon_0, \varepsilon)^{2/3}/T^{1/3}$ where $\varepsilon_0 = T^{-1/4} \log^{3/4} d$, the $\mathsf{L2P}$ transformation (Algorithm 1) applied with the measure $\{\mu_t^{\mathsf{mw}}\}_{t=1}^T$ is $(\varepsilon, \delta)$-DP and has regret*

$$\mathbf{Reg}_T = O\left(\sqrt{T \log d} + \frac{T^{1/3} \log d}{\varepsilon^{2/3}}\right).$$

*Proof.* First, based on theorem 3.2, note that the setting of $B = 1/\varepsilon$ and $\eta \leq \min(\varepsilon_0, \varepsilon)^{2/3}/T^{1/3}$ where $\varepsilon_0 = T^{-1/4} \log^{3/4} d$ guarantee the algorithm is $(\varepsilon, \delta)$-DP.

To upper bound the regret, we use existing guarantees of the multiplicative weights algorithm [AHK12], combined with Theorem 3.2 to get that the regret is

$$\mathbf{Reg}_T \leq O\left(\eta T + \frac{\log(d)}{\eta} + TB^2\eta^2\right)$$

$$\leq O\left(\eta T + \frac{\log(d)}{\eta} + \frac{T\eta^2}{\varepsilon^2}\right)$$

$$\leq O\left((T\varepsilon_0)^{2/3} + \frac{T^{1/3}\log(d)}{\varepsilon^{2/3}} + \frac{T^{1/3}}{\varepsilon^{2/3}}\right)$$

$$\leq O\left(\sqrt{T\log d} + \frac{T^{1/3}\log(d)}{\varepsilon^{2/3}}\right),$$

where the second inequality follows by setting $B = 1/\varepsilon$, and the third inequality follows by setting $\eta \leq \min(\varepsilon_0, \varepsilon)^{2/3}/T^{1/3}$, and the last inequality follows since $\varepsilon_0 = T^{-1/4} \log^{3/4} d$. $\qquad\square$

### 3.2 Application to DP-OCO

In this section, we use our transformation for differentially private online convex optimization (DP-OCO) using the regularized multiplicative weights algorithm [AKST23b], which has the following measure

$$\mu_t^{\mathsf{rmw}}(x) = e^{-\beta\left(\sum_{i-1}^{t-1} \ell_i(x) + \lambda\|x\|_2^2\right)}. \tag{3}$$

Letting $\overline{\mu}^{\mathsf{rmw}}$ denote the corresponding density function, we have the following properties.

**Lemma 3.11.** *Assume $\ell_1, \ldots, \ell_T : \mathcal{X} \to \mathbb{R}$ be convex and $L$-Lipschitz functions. Then we have that*

1. $D_\infty^{\delta_0}(\overline{\mu}_{t+1}^{\mathsf{rmw}}, \overline{\mu}_t^{\mathsf{rmw}}) \leq \eta$ *where* $\eta = \frac{2\beta L^2}{\lambda} + \sqrt{\frac{8\beta L^2 \log(2/\delta_0)}{\lambda}}$.

2. $\frac{\mu_{t+1}^{\mathsf{rmw}}(x)}{\mu_t^{\mathsf{rmw}}(x)} = e^{-\beta\ell_t(x)}$ *for all $x \in \mathcal{X}$.*

*Proof.* The first item follows from Lemma 3.5 in [GLL22, AKST23b]. The second item follows immediately from the definition of $\mu_t^{\mathsf{rmw}}$. $\qquad\square$

Combining these properties with our transformation, we get the following result.

**Theorem 3.12** (DP-OCO). *Let $\ell_1, \ldots, \ell_T : \mathcal{X} \to \mathbb{R}$ be convex and $L$-Lipschitz functions. Setting $B = \frac{1}{2\varepsilon \log(1/\delta)}$, $\lambda = \frac{L}{D}\max\{\sqrt{T}, \frac{\sqrt{d\log T}}{\eta}\}$, $\beta = \eta^2\lambda/20L^2$, $\eta = \frac{\varepsilon^{2/3}}{T^{1/3}\log(T/\delta)}$ and $p = \eta/\varepsilon$, the L2P transformation (Algorithm 1) applied with the measure $\{\mu_t^{\mathsf{rmw}}\}_{t=1}^T$ is $(\varepsilon, \delta)$-DP and has regret*

$$\mathbf{Reg}_T = LD \cdot O\left(\sqrt{T} + \frac{T^{1/3}\sqrt{d\log T}\log(T/\delta)}{\varepsilon^{2/3}}\right).$$

*Proof.* First, based on Theorem 3.2, note that there are three constraints to make the algorithm private:

$$\eta/p \leq \varepsilon/2, \qquad \eta\sqrt{Tp\log(1/\delta)/B} \leq \varepsilon/2, \qquad \eta B\log(1/\delta)/p \leq 1.$$

Setting of $B = \frac{1}{2\varepsilon\log(1/\delta)}$, $\lambda = \frac{L}{D}\max\{\sqrt{T}, \frac{\sqrt{d\log T}}{\eta}\}$, $\beta = \eta^2\lambda/20L^2$, $\eta = \frac{\varepsilon^{2/3}}{T^{1/3}\log(T/\delta)}$ and $p = \eta/\varepsilon$ guarantees the algorithm is $(\varepsilon, \delta)$-DP.

For utility, we use theorem 3.2 with the existing regret bounds for the regularized multiplicative weights algorithm (Theorem 4.1 in [AKST23b]) to get that the algorithm has regret

$$\mathbf{Reg}_T \leq O\left(\lambda D^2 + \frac{L^2T}{\lambda} + \frac{d\log(T)}{\beta} + LDTB^2\eta^2\right)$$

$$\leq O\left( LD\sqrt{T} + \lambda D^2 + \frac{L^2 d \log T}{\lambda \eta^2} + LDTB^2\eta^2 \right)$$

$$\leq LD \cdot O\left( \sqrt{T} + \frac{T^{1/3}\sqrt{d \log T}\log(T/\delta)}{\varepsilon^{2/3}} \right).$$

$\square$

## 4   Lower bound for low-switching private algorithms

In this section, we prove a lower bound for DP-OPE for a natural family of private low-switching algorithms that contains most of the existing low-switching private algorithms such as our algorithms and the ones in [AFKT23b, AKST23b]. Our lower bound matches our upper bounds for DP-OPE and suggests that new techniques beyond limited switching are required in order to obtain faster rates.

For our lower bounds, we will assume that the algorithm satisfies the following condition:

**Condition 4.1.** *(Limited switching algorithms) The online algorithm* ALG *works as follows: at each round t,* ALG *is allowed to either set $x_{t+1} = x_t$ or sample $x_{t+1} \sim \mu_{t+1}$ where $\mu_{t+1}$ is a function of $\ell_1, \ldots, \ell_t$ and is supported over $\mathcal{X}$. The algorithm releases the resampling rounds $\{t_1, \ldots, t_S\}$ and models $\{x_{t_1}, \ldots, x_{t_S}\}$.*

Our lower bound will hold for algorithms that satisfy concentrated differential privacy. We use this notion as it allows to get tight characterization of the composition of private algorithms and in most settings have similar rates to approximate differential privacy. We can also prove a tight lower bound for pure differential privacy using the same techniques. We have the following lower bound for concentrated DP. We defer the proof to Appendix C.

**Theorem 4.2.** *Let $T \geq 1$ and $\varepsilon \geq 100\log^{3/2}(dT)/T$. If an algorithm* ALG *satisfies Condition 4.1 and is $\varepsilon^2$-CDP, then there exists an oblivious adversary that chooses $\ell_1, \ldots, \ell_T : [d] \to [0, 1]$ such that the regret is lower bounded by*

$$\mathbf{Reg}_T \geq \Omega\left( \sqrt{T} + \frac{T^{1/3}}{\varepsilon^{2/3}} \right).$$

Finally, we note that this lower bound only holds for switching-based algorithms: indeed, the binary-tree-based algorithm of [AS17] obtains regret $\sqrt{d}\log(d)/\varepsilon$ which is better in the low-dimensional regime. This motivates the search for new strategies beyond limited switching for the high-dimensional regime.

## 5   Conclusion

In this paper, we proposed a new transformation that allows the conversion of lazy online learning algorithms into private algorithms and demonstrates two applications (DP-OPE and DP-OCO) where this transformation offers significant improvements over prior work. Moreover, for DP-OPE, we show a lower bound for natural low-switching-based private algorithms, which shows that new techniques are required for low-switching algorithms to improve our transformation's regret. This begs the question of whether the same lower bound holds for all algorithms or whether a different strategy that breaks the low-switching lower bound exists. As for DP-OCO, it is interesting to see whether better upper or lower bounds can be obtained. The current normalized regret, omitting logarithmic terms, is proportional to $\sqrt{d}/(\varepsilon T)^{2/3}$. This is different than most applications in private optimization where the normalized error is usually a function of $\sqrt{d}/(\varepsilon T)$. Hence, it is natural to conjecture that the normalized regret can be improved to $d^{1/3}/(\varepsilon T)^{2/3}$.

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

# A  Missing details for preliminaries

In this section, we provide additional preliminaries and provide missing details for some of the results in the preliminaries section.

For our lower bounds, we require the notion of concentrated differential privacy. To this end, we first define the $\alpha$-Renyi divergence ($\alpha > 1$) between two probability measures:

$$D_\alpha(\mu\|\nu) := \frac{1}{\alpha - 1} \log \left( \int \left( \frac{\mu(\omega)}{\nu(\omega)} \right)^\alpha d\nu(\omega) \right).$$

Concentrated DP is defined below:

**Definition A.1** (concentrated DP). Let $\rho \geq 0$. We say an algorithm ALG satisfies $\rho$-concentrated differential privacy ($\rho$-CDP) against oblivious adversaries if for any neighboring sequences $\mathcal{S} = (\ell_1, \ldots, \ell_T) \in \mathcal{L}^T$ and $\mathcal{S}' = (\ell'_1, \ldots, \ell'_T) \in \mathcal{L}^T$ that differ in a single element, and any $\alpha \geq 1$, $D_\alpha(\mathsf{ALG}(\mathcal{D})\|\mathsf{ALG}(\mathcal{D}')) \leq \alpha\rho$.

Now we list a few standard results from the privacy literature that we use in the paper, namely group privacy and privacy composition.

**Lemma A.2.** *(Group Privacy) Let* ALG *be an* $(\varepsilon, \delta)$*-DP algorithm and let* $\mathcal{S}, \mathcal{S}'\mathcal{L}^T$ *be two datasets that differ in* $k$ *elements. Then for any measurable set* $S$ *in the output space of* ALG

$$\Pr[\mathsf{ALG}(\mathcal{S}) \in \mathcal{O}] \leq e^{k\varepsilon} \Pr[\mathsf{ALG}(\mathcal{S}') \in \mathcal{O}] + ke^{(k-1)\varepsilon}\delta.$$

**Lemma A.3** (Advanced Composition,[KOV15]). *For any* $\varepsilon_t > 0, \delta_t \in (0, 1)$ *for* $t \in [k]$, *and* $\tilde{\delta} \in (0, 1)$, *the class of* $(\varepsilon_t, \delta_t)$*-DP mechanisms satisfy* $(\tilde{\varepsilon}_{\tilde{\delta}}, 1 - (1 - \tilde{\delta})\Pi_{t\in[k]}(1 - \tilde{\delta}_t))$*-DP under* $k$*-fold adaptive composition, for*

$$\tilde{\varepsilon}_{\tilde{\delta}} = \sum_{t\in[k]} \varepsilon_t + \min \left\{ \sqrt{\sum_{t\in[k]} 2\varepsilon_t^2 \log(e + \frac{\sqrt{\sum_{t\in[k]} \varepsilon_t^2}}{\tilde{\delta}})}, \sqrt{\sum_{t\in[k]} 2\varepsilon_t^2 \log(1/\tilde{\delta})} \right\}. \tag{4}$$

Moreover, we use the fact that distributions with bounded $D_\infty^\delta$ satisfy the following property.

**Lemma A.4.** *Let* $\varepsilon \leq 1/10$. *If* $D_\infty^\delta(\mu, \nu) \leq \varepsilon/2$ *then we have*

$$\Pr_{X\sim\mu} \left[ e^{-\varepsilon} \leq \frac{\mu(X)}{\nu(X)} \leq e^\varepsilon \right] \geq 1 - 6\delta/\varepsilon \text{ and } \Pr_{X\sim\nu} \left[ e^{-\varepsilon} \leq \frac{\mu(X)}{\nu(X)} \leq e^\varepsilon \right] \geq 1 - 6\delta/\varepsilon.$$

Finally, we have the following standard conversion from $\rho$-concentrated DP to $(\varepsilon, \delta)$-DP.

**Lemma A.5** ([BS16]). *If* ALG *is* $\rho$*-CDP with* $\rho \leq 1$, *then it is* $(3\sqrt{\rho \log(1/\delta)}, \delta)$*-DP for all* $\delta \in (0, 1/4)$.

## A.1  Proof of Lemma A.4

Let $S' = \{X : \frac{\mu(X)}{\nu(X)} \in [e^{-\varepsilon}, e^\varepsilon]\}$, and assume towards a contradiction that $\mu(S') < 1 - 6\delta/\varepsilon$. Then there is a set $S = \overline{S'}$, such that $\mu[S] > 6\delta/\varepsilon$. We divide $S$ by letting $S_1 = \{X \in S : \mu(X)/\nu(X) > e^\varepsilon\}$ and $S_2 = S \setminus S_1$.

**Case (1):** $\mu(S_1) \geq \mu(S)/2 \geq 3\delta/\varepsilon$. Then we know

$$\ln \frac{\mu(S_1) - \delta}{\nu(S_1)} > \ln e^\varepsilon \frac{\mu(S_1) - \delta}{\mu(S_1)} \geq \ln e^\varepsilon \frac{3/\varepsilon - 1}{3/\varepsilon} \geq \ln e^{\varepsilon/2} = \varepsilon/2,$$

where we use that $\mu(S_1) \geq e^\varepsilon\nu(S_1)$ and $1 - \varepsilon/3 \geq e^{-\varepsilon/2}$ for $\varepsilon < 1/10$. This is a contradiction.

**Case (2):** Suppose $\mu(S_2) \geq 3\delta/\varepsilon$. We know $S_2 = \{X \in S : \mu(X)/\nu(X) < e^{-\varepsilon}\} = \{X \in S : \nu(X)/\mu(X) > e^\varepsilon\}$. Then we know $\nu(S_2) \geq e^\varepsilon\mu(S_2) \geq 3e^\varepsilon\delta/\varepsilon$. Similarly, we have

$$\ln \frac{\nu(S_2) - \delta}{\mu(S_2)} > \varepsilon/2.$$

This is a contradiction as well and proves the statement.

# B  Missing Proofs for Section 3

## B.1  Proof of Theorem 3.2

The regret bound follows directly from Proposition 3.3. It suffices to prove the privacy guarantee.

Fix two arbitrary neighboring datasets $\mathcal{S}$ and $\mathcal{S}'$, and suppose the sequences differ at $t_0$-step, that is $\{\ell_{(t_0-1)B+1}, \cdots, \ell_{t_0 B}\}$ differ one loss function from $\{\ell'_{(t_0-1)B+1}, \cdots, \ell'_{t_0 B}\}$.

Define $\zeta_t$ as the indicator that at least one of $A_{t+1}, S_{t+1}$ and $S'_{t+1}$ is zero. Let $\{(x_t, y_t, \zeta_t)\}_{t \in [T]}$ and $\{(x'_t, y'_t, \zeta'_t)\}_{t \in [T]}$ be the random variables with neighboring datasets. Let $\Sigma_t = \{(x_\tau, y_\tau, \zeta_\tau)\}_{\tau \in [t]}$ be the random variables for the first $t$-iterations. We will argue that $\{(x_t, y_t, \zeta_t)\}_{t \in [T]}$ and $\{(x'_t, y'_t, \zeta'_t)\}_{t \in [T]}$ are indistinguishable, and privacy will follow immediately.

Let $E_t$ be the event such that $\frac{\nu_{t+1}(x_t)}{\nu_t(x_t)} \cdot \frac{\nu_t(y_t)}{\nu_{t+1}(y_t)} \in [e^{-2B\eta}, e^{2B\eta}]$. Hence we know $\Pr[E_t] \geq 1 - (2 + \log(1/\delta_1)/p)eB\delta_0 - \delta_1$ by Lemma 3.7 for any $t$. Define $E'_t$ in a similar way. Moreover, let $E_G$ be the event that $\sum_{t=2}^{T/B} \mathbf{1}(A_t = 0 \quad \text{or} \quad S'_t = 0) \leq 2Tp\log(1/\delta_1)/B$. By Chernoff bound, we know

$$\Pr(E_G) = \Pr[\sum_{t=2}^{T/B} \mathbf{1}(A_t = 0 \quad \text{or} \quad S'_t = 0) \leq 2Tp\log(1/\delta_1)/B] \geq 1 - \delta_1.$$

We also let $E_t$ be the event such that $\frac{\nu_{t+1}(x_t)}{\nu_t(x_t)} \cdot \frac{\nu_t(y_t)}{\nu_{t+1}(y_t)} \in [e^{-2B\eta}, e^{2B\eta}]$. Lemma 3.7 implies that $\Pr[E_t] \geq 1 - (2/\eta + \log(1/\delta_1)/p)eB\delta_0 - \delta_1$ for any $t$. Define $E'_t$ in a similar way.

Then it suffices to show $\{(x_t, y_t, \zeta_t)\}_{t \in [T]}$ and $\{(x'_t, y'_t, \zeta'_t)\}_{t \in [T]}$ are $(\varepsilon, \delta_x)$-indistinguishable conditional on $E := E_G \cup E'_G \cup_{t \in [T/B]} (E_t \cup E'_t)$. Then this will imply that $\{(x_t, y_t, \zeta_t)\}_{t \in [T]}$ and $\{(x'_t, y'_t, \zeta'_t)\}_{t \in [T]}$ are $(\varepsilon, \delta_x + (2T+2)\delta_1 + 2T(2/\eta + \log(1/\delta_1)/p)eB\delta_0)$-indistinguishable.

Now we show, conditional on $E$, any value $\Sigma$ such that $\Sigma_{t-1} = \Sigma'_{t-1} = \Sigma$, $(x_t, y_t, \zeta_t)$ and $(x'_t, y'_t, \zeta'_t)$ are $(\varepsilon_t, \delta_0)$-indistinguishable where

$$\varepsilon_t = \begin{cases} 0, & t < t_0 \\ 2\eta/p & t = t_0 \\ \zeta_t \cdot \eta & t > t_0 \end{cases} \tag{5}$$

**Case 1** ($t < t_0$)**:**  It is clear that the claim is correct for $t \leq t_0$ as $(x_t, y_t, \zeta_t)$ and $(x'_t, y'_t, \zeta'_t)$ have the same distribution then.

**Case 2** ($t = t_0$)**:**  Now consider the case where $t = t_0$.

Note that $(x_{t_0}, y_{t_0})$ and $(x'_{t_0}, y'_{t_0})$ have identical distributions, and it suffices to consider the indistinguishability of $\zeta_{t_0}$ and $\zeta'_{t_0}$.

We have

$$\frac{\Pr[\zeta_{t_0} = 0 \mid \Sigma_{t_0-1}, x_{t_0}, y_{t_0}]}{\Pr[\zeta'_{t_0} = 0 \mid \Sigma'_{t_0-1}, x'_{t_0}, y'_{t_0}]} = \frac{(1-p)^2 \frac{\nu_{t_0+1}(x_{t_0})}{e^{2B\eta}\nu_{t_0}(x_{t_0})} \cdot \frac{\nu_{t_0}(y_{t_0})}{\nu_{t_0+1}(y_{t_0})}}{(1-p)^2 \frac{\nu'_{t_0+1}(x'_{t_0})}{e^{2B\eta}\nu'_{t_0}(x'_{t_0})} \cdot \frac{\nu'_{t_0}(y'_{t_0})}{\nu'_{t_0+1}(y'_{t_0})}}$$

$$= \frac{\nu_{t_0+1}(x_{t_0})}{\nu'_{t_0+1}(x_{t_0})} \cdot \frac{\nu'_{t_0+1}(y_{t_0})}{\nu_{t_0+1}(y_{t_0})}$$

$$\leq e^{2\eta}.$$

Similarly, we have

$$\frac{\Pr[\zeta_{t_0} = 1 \mid \Sigma_{t_0-1}, x_{t_0}, y_{t_0}]}{\Pr[\zeta'_{t_0} = 1 \mid \Sigma'_{t_0-1}, x'_{t_0}, y'_{t_0}]} = \frac{1 - (1-p)^2 + (1-p)^2(1 - \frac{\nu_{t_0+1}(x_{t_0})}{e^{2B\eta}\nu_{t_0}(x_{t_0})} \frac{\nu_{t_0}(y_{t_0})}{\nu_{t_0+1}(y_{t_0})})}{1 - (1-p)^2 + (1-p)^2(1 - \frac{\nu'_{t_0+1}(x'_{t_0})}{e^{2B\eta}\nu'_{t_0}(x'_{t_0})} \frac{\nu'_{t_0}(y'_{t_0})}{\nu'_{t_0+1}(y'_{t_0})})}$$

$$= 1 + \frac{(1-p)^2 \left( \frac{\nu'_{t_0+1}(x'_{t_0})}{e^{2B\eta}\nu'_{t_0}(x'_{t_0})} \frac{\nu'_{t_0}(y'_{t_0})}{\nu'_{t_0+1}(y'_{t_0})} - \frac{\nu_{t_0+1}(x_{t_0})}{e^{2B\eta}\nu_{t_0}(x_{t_0})} \frac{\nu_{t_0}(y_{t_0})}{\nu_{t_0+1}(y_{t_0})} \right)}{1 - (1-p)^2 + (1-p)^2 \left( 1 - \frac{\nu'_{t_0+1}(x'_{t_0})}{e^{2B\eta}\nu'_{t_0}(x'_{t_0})} \frac{\nu'_{t_0}(y'_{t_0})}{\nu'_{t_0+1}(y'_{t_0})} \right)}$$

$$\leq 1 + \frac{e^{2\eta} - 1}{p} \leq e^{2\eta/p}.$$

**Case 3** $(t > t_0)$: As for the case when $t > t_0$, when $\zeta_t = 0$ ($A_{t+1} = S_{t+1} = S'_{t+1} = 1$), the variables are 0-indistinguishable since $x_t = x_{t-1}$ and $y_t = y_{t-1}$ in this case. Consider the remaining possibility. Given the assumption that $\mu_{t+1}/\mu_t$ is a function of $\ell_t$, for any possible $\Sigma$, we have

$$\Pr[\zeta_t = 1 \mid \Sigma_{t-1} = \Sigma] = \Pr[\zeta'_t = 1 \mid \Sigma'_{t-1} = \Sigma].$$

For any set $S$, by the assumption on $\overline{\mu}_t$, we have

$$\begin{aligned}
&\Pr[\zeta_t = 1, (x_t, y_t) \in S \mid \Sigma_{t-1} = \Sigma] \\
&= \Pr[(x_t, y_t) \in S \mid \Sigma_{t-1} = \Sigma, \zeta_t = 1] \Pr[\zeta_t = 1 \mid \Sigma_{t-1} = \Sigma] \\
&= \Pr[(x_t, y_t) \in S \mid \Sigma_{t-1} = \Sigma, \zeta_t = 1] \Pr[\zeta'_t = 1 \mid \Sigma'_{t-1} = \Sigma] \\
&\leq e^{2\eta} \Pr[(x'_t, y'_t) \in S \mid \Sigma'_{t-1} = \Sigma, \zeta'_t = 1] \Pr[\zeta'_t = 1 \mid \Sigma'_{t-1} = \Sigma] + 4\delta_0 \\
&= e^{2\eta} \Pr[\zeta'_t = 1, (x'_t, y'_t) \in S \mid \Sigma'_{t-1} = \Sigma] + 4\delta_0,
\end{aligned}$$

where the inequality comes from Lemma 3.6 by the divergence bound between $\overline{\mu}_t$ and $\overline{\mu}'_t$. This completes the proof of Equation (5).

The final privacy guarantee follows from combining Equation (5) and Advanced composition (Lemma A.3).

## B.2  Proof of Lemma 3.4

We prove this statement by induction. For $t = 1$, the statement is obviously correct. We assume $\|\widehat{\nu}_t - \overline{\nu}_t\|_{TV} \leq 3(t-1)(2(e/\eta + \log(1/\delta_1)/p)B\delta_0 + \delta_1)$ prove that $\|\widehat{\nu}_{t+1} - \overline{\nu}_{t+1}\|_{TV} \leq 3t(2e + \log(1/\delta_0)/p)B\delta_0$.

Let $X_{good} := \{x : \log \frac{\overline{\nu}_{t+1}(x)}{\overline{\nu}_t(x)} \in [-B\eta, B\eta]\}$ and $Y_{good} := \{y : \log \frac{\overline{\nu}_t(y)}{\overline{\nu}_{t+1}(y)} \leq [-B\eta, B\eta]\}$. Let $\widehat{\varphi}_t(y)$ be the distribution of $y_t$. Note that the distribution of $y_t$ is independent of $\{x_\tau\}_{\tau \in [T/B]}$, while the distribution of $x_{t+1}$ is independent of $y_{t+1}$ but depends on $y_t$. By the assumption and group privacy, we know $D_\infty^{Be^{B\eta}\delta_0}(\overline{\nu}_{t+1}, \overline{\nu}_t) \leq B\eta$, and hence we have

$$\nu_t(Y_{good}^{\complement}) \leq e^{B\eta}\delta_0/\eta \leq 2e\delta_0/\eta.$$

Let $t_0 \leq t$ be largest integer such that $A_{t_0} = 1$, that is, $y_t$ is sampled from $\overline{\nu}_{t_0}$ for some random $t_0 \leq t$. We have

$$\nu_{t_0}(Y_{good}^{\complement}) \leq e^{B\eta(t-t_0)} \cdot \nu_t(Y_{good}) + (t - t_0)B\delta_0 e^{B\eta(t-t_0)}.$$

With probability at least $1 - \delta_1$, we know $|t - t_0| \leq \log(1/\delta_1)/p$. Hence we get

$$\Pr_{y \sim \widehat{\varphi}_t}[y \in Y_{good}] \geq 1 - 2(e/\eta + \log(1/\delta_1)/p)B\delta_0 - \delta_1.$$

We know

$$\begin{aligned}
&\Pr_{x \sim \overline{\nu}_t, y \sim \widehat{\varphi}_t}[x \in X_{good}, y \in Y_{good}] \\
&= \Pr_{y \sim \widehat{\varphi}_t}[y \in Y_{good}] \Pr_{x \sim \overline{\nu}_t}[x \in X_{good} \mid y \in Y_{good}] \\
&\geq 1 - 2(e/\eta + \log(1/\delta_1)/p)B\delta_0 - \delta_1.
\end{aligned}$$

Denote the good set

$$S_{good} = \big\{(x, y) : x \in X_{good}, Y_{good}\big\}.$$

Let $\tilde{\varphi}_t$ be the distribution of $y_t$ conditional on $y_t \in Y_{good}$. Let $\widehat{\Gamma}_t$ be the marginal distribution over $(x_t, y_t)$, that is $x_t \sim \widehat{\nu}_t$ and $y_t \sim \widehat{\varphi}_t$. Let $\Gamma_t$ be the distribution over $(x_t, y_t)$ where $x_t \sim \overline{\nu}_t, y_t \sim \tilde{\varphi}_t$, and $\overline{\Gamma}_t$ be the distribution of $\Gamma_t$ conditional on $(x_t, y_t) \in S_{good}$.

We know $\|\widehat{\Gamma}_t - \overline{\Gamma}_t\|_{TV} \leq (2e/\eta + \log(1/\delta_1)/p)B\delta_0(3t - 2)$. Let $\overline{q}_{t+1}$ be the distribution of $x_{t+1}$ if $(x_t, y_t)$ is sampled from $\overline{\Gamma}_t$ instead of $\widehat{\Gamma}_t$. By the property that post-processing does not increase the TV distance, we know

$$\|\overline{q}_{t+1} - \widehat{\nu}_{t+1}\|_{TV} \leq \|\widehat{\Gamma}_t - \overline{\Gamma}_t\|_{TV}.$$

Now it suffices to bound the TV distance between $\overline{q}_{t+1}$ and $\overline{\nu}_{t+1}$.

For any set $E$, we have

$$
\begin{aligned}
\overline{q}_{t+1}(E) = \int & (\Pr[S'_t = 0, x_{t+1} \in E \mid x_t = x, y_t = y] \\
& + \Pr[S'_t = 1, S_t = 0, x_{t+1} \in E \mid x_t = x, y_t = y] \\
& + \Pr[S'_t = 1, S_t = 1, x_{t+1} \in E \mid x_t = x, y_t = y])\overline{\Gamma}_t(x, y)\mathrm{d}(x, y) \\
= & \, p\overline{\nu}_{t+1}(E) + (1-p)\overline{\nu}_{t+1}(E)\int (1 - \frac{\nu_{t+1}(x)}{e^{2B\eta}\nu_t(x)} \cdot \frac{\nu_t(y)}{\nu_{t+1}(y)})\overline{\Gamma}_t(x, y)\mathrm{d}(x, y) \\
& + (1-p)\int_{\mathbf{1}(x\in E)} \frac{\nu_{t+1}(x)}{e^{2B\eta}\nu_t(x)} \cdot \frac{\nu_t(y)}{\nu_{t+1}(y)}\overline{\Gamma}_t(x, y)\mathrm{d}(x, y).
\end{aligned}
$$

Thus we have

$$
\begin{aligned}
|\overline{q}_{t+1}(E) - \overline{\nu}_{t+1}(E)| \leq \Big| & \int_{\mathbf{1}(x\in E)} \frac{\overline{\nu}_{t+1}(x)}{e^{2B\eta}\overline{\nu}_t(x)} \cdot \frac{\overline{\nu}_t(y)}{\overline{\nu}_{t+1}(y)}\overline{\Gamma}_t(x, y)\mathrm{d}(x, y) \\
& - \overline{\nu}_{t+1}(E)\int \frac{\overline{\nu}_{t+1}(x)}{e^{2B\eta}\overline{\nu}_t(x)} \cdot \frac{\overline{\nu}_t(y)}{\overline{\nu}_{t+1}(y)}\overline{\Gamma}_t(x, y)\mathrm{d}(x, y)\Big|.
\end{aligned}
$$

Note that for any $(x, y) \in S_{good}$, we have

$$\overline{\Gamma}_t(x, y) = \frac{\overline{\nu}_t(x)\tilde{\varphi}_t(y)}{\Gamma_t(S_{good})}.$$

Fixing any $y$, we know the above term is bounded by

$$|\frac{\overline{\nu}_t(y)}{e^{2B\eta}\overline{\nu}_{t+1}(y)\Gamma_t(S_{good})}(\overline{\nu}_{t+1}(E \cap X_{good}) - \overline{\nu}_{t+1}(E)\overline{\nu}_{t+1}(X_{good}))| \leq 2(e/\eta + B\log(1/\delta_1)/p)\delta_0,$$

where the last inequality follows from $\overline{\nu}_{t+1}(X_{good}) \geq 1 - B\delta_0$. Hence, we prove that

$$\|\overline{q}_{t+1} - \overline{\nu}_{t+1}\|_{TV} \leq 2(e/\eta + \log(1/\delta_1)/p)B\delta_0.$$

This suggests that

$$\|\widehat{\nu}_{t+1} - \overline{\nu}_{t+1}\|_{TV} \leq \|\widehat{\nu}_{t+1} - \overline{q}_{t+1}\|_{TV} + \|\overline{q}_{t+1} - \overline{\nu}_{t+1}\|_{TV} \leq 6t(e/\eta + \log(1/\delta_1)/p)B\delta_0.$$

## B.3 Proof of Lemma 3.6

Let $\mathcal{S} = (\ell_1, \ldots, \ell_T)$ and $\mathcal{S}' = (\ell'_1, \ldots, \ell'_T)$ differ in a single round $t_0$. We fix $t$ and prove the claim is correct. If $t \leq t_0$, then the claim clearly holds as $\overline{\mu}_t = \overline{\mu}'_t$. For $t = t_0 + 1$, note that Assumption 3.1 implies that $D^{\delta_0}_\infty(\overline{\mu}_{t_0+1}, \overline{\mu}_{t_0}) \leq \eta$ and $D^{\delta_0}_\infty(\overline{\mu}'_{t_0+1}, \overline{\mu}_{t_0}) \leq \eta$, hence by group privacy we get that $D^{(e^\eta+1)\delta_0}_\infty(\overline{\mu}_t, \overline{\mu}'_t) \leq 2\eta$. Finally, for $t > t_0 + 1$, note that Assumption 3.1 implies that $\mu_t = \mu_0 \cdot \mathsf{func}(\ell_1) \cdot \mathsf{func}(\ell_2) \cdots \mathsf{func}(\ell_{t-1})$ and $\mu'_t = \mu_0 \cdot \mathsf{func}(\ell'_1) \cdot \mathsf{func}(\ell'_2) \cdots \mathsf{func}(\ell'_{t-1})$. Thus, swapping the losses at rounds $t - 1$ and $t_0$ results in the same distributions $\mu_t$ and $\mu'_t$, therefore privacy follows from the same arguments as the case when $t = t_0 + 1$. The final claim follows as $e^\eta + 1 \leq 4$.

## B.4 Proof of Lemma 3.7

To prove lemma 3.7, we first prove the same result under a simpler setting where $x_t \sim \nu_t$ and $y_t \sim \nu_t$.

**Lemma B.1.** *For any $0 \leq t \leq T/B - 1$, if $B\eta \leq 1/20$, $x_t \sim \nu_t$ and $y_t \sim \nu_t$ independently, then with probability at least $1 - 6e^{B\eta}\delta_0/\eta$,*

$$\frac{\nu_{t+1}(x_t)}{\nu_t(x_t)} \cdot \frac{\nu_t(y_t)}{\nu_{t+1}(y_t)} \in [e^{-2B\eta}, e^{2B\eta}]$$

*Proof.* Let $Z_t = \int \nu_t(x)\mathrm{d}x$. We know $\overline{\nu}_t = \nu_t/Z_t$ by our notation. Then we have that

$$\frac{\nu_{t+1}(x_t)}{\nu_t(x_t)} \cdot \frac{\nu_t(y_t)}{\nu_{t+1}(y_t)} = \frac{\nu_{t+1}(x_t)Z_t}{\nu_t(x_t)Z_{t+1}} \cdot \frac{\nu_t(y_t)Z_{t+1}}{\nu_{t+1}(y_t)Z_t}$$

$$= \frac{\overline{\nu}_{t+1}(x_t)}{\overline{\nu}_t(x_t)} \cdot \frac{\overline{\nu}_t(y_t)}{\overline{\nu}_{t+1}(y_t)}.$$

The statement follows from the Assumption 3.1 and the group privacy

$$D_\infty^{Be^{B\eta}\delta_0}(\overline{\nu}_{t+1}, \overline{\nu}_t) \leq B\eta.$$

Then the statement follows from Lemma A.4, the independence between $x_t, y_t$ and Union bound. $\quad\square$

We are now ready to prove Lemma 3.7.

*Proof.* Fix any $t$. Let $t_0 \leq t$ be largest integer such that $A_{t_0} = 1$, that is, $y_t$ is sampled from $\overline{\nu}_{t_0}$ for some random $t_0 \leq t$. By the group privacy, we know $D_\infty^{Be^{B\eta(t-t_0)}\delta_0(t-t_0)}(\nu_t, \nu_{t_0}) \leq B\eta(t-t_0)$.
Define the bad set

$$S_{bad} = \{y : \frac{\nu_{t+1}(x)}{\nu_t(x)} \cdot \frac{\nu_t(y)}{\nu_{t+1}(y)} \notin [e^{-2B\eta}, e^{2B\eta}], x \sim \nu_t\}.$$

By Lemma B.1, we know

$$\nu_t(y \in S_{bad}) \leq 6e^{B\eta} \cdot \delta_0/\eta.$$

Therefore, we have that

$$\nu_{t_0}(y \in S_{bad}) \leq e^{B\eta(t-t_0)} \cdot \nu_t(y \in S_{bad}) + (t-t_0)B\delta_0 e^{B\eta(t-t_0)}$$

$$\leq 2e^{B\eta(t-t_0+1)} \cdot B\delta_0/\eta + B\delta_0(t-t_0)e^{B\eta(t-t_0)}.$$

By the CDF of the geometric distribution, we know with probability at least $1 - \delta_1$, we get $|t_0 - t| \leq \log(1/\delta_1)/p$. Let $E$ be the event that $|t_0 - t| \leq \log(1/\delta_1)/p$. Hence we know

$$\nu_{t_0}(y \in S_{bad}) \leq \nu_{t_0}(y \in S_{bad} \mid E)\Pr(E) + \Pr(E^c)$$

$$\leq (2/\eta + \log(1/\delta_1)/p) \cdot eB\delta_0 + \delta_1.$$

$\quad\square$

## C   Missing proofs for Section 4

We prove a sequence of lemmas that are needed for the proof. The first lemma shows that the algorithm has to split the privacy budget across all resampling rounds. To this end, let $S$ be a random variable that corresponds to the number of resampling steps in the algorithm, let $T_i$ be the random variable corresponding to the round of the $i$'th resampling (where we let $T_i = T + 1$ if $i > S$), and let $Z_i$ be the random variable corresponding to the model sampled at time $T_i$ (letting $Z_i = 1$ if $i > S$).

**Lemma C.1.** *(Composition)  Let $S, T_i, Z_i$ and $S', T_i', Z_i'$ denote the random variables for two neighboring datasets. Under the assumptions of Theorem 4.2, if* ALG *is $\varepsilon^2$-CDP, then for all $\alpha \geq 1$*

$$\sum_{i=1}^T D_\alpha(Z_i || Z_i' \mid T_i) \leq \alpha\varepsilon^2.$$

*Proof.* As ALG is $\varepsilon^2$-concentrated DP and outputs $T_1, \ldots, T_S$ and $Z_1, \ldots, Z_S$, we have that

$$
\begin{aligned}
\alpha\varepsilon^2 &\geq D_\alpha(T_1, Z_1, \ldots, T_S, Z_S \| T_1', Z_1', \ldots, T_{S'}', Z_{S'}') \\
&\geq D_\alpha(T_1, Z_1, \ldots, T_T, Z_T \| T_1', Z_1', \ldots, T_T', Z_T') \\
&\geq \sum_{i=1}^{T} D_\alpha(Z_i \| Z_i' \mid T_i),
\end{aligned}
$$

where the second inequality follows as the random variables $T_i, Z_i$ and $T_j', Z_j'$ are constant for $i > S$ and $j > S'$, and the last inequality follows as $Z_i$ is independent of $(T_1, \ldots, T_i)$ and $(Z_1, \ldots, Z_{i-1})$ given $T_i$. $\qquad\square$

We defer the proof of the following Lemma to the appendix.

**Lemma C.2.** *Let $T \geq 1$, $\varepsilon \leq 1/T$ and $\delta \leq 1/2$. Assume $\ell : [d] \to \{0, 1\}$ where $\ell[x] \sim \mathsf{Ber}(1/2)$ for each $x \in [d]$. Let $D = (\ell, \ldots, \ell)$ and let $\mathsf{ALG}$ be an $(\varepsilon, \delta)$-DP algorithm that outputs $(z_1, \ldots, z_T) = \mathsf{ALG}(D)$. Then*

$$
\mathbb{E}[\sum_{t=1}^{T} \ell(z_t)] \geq T \cdot \left( \frac{1}{2} - \frac{T\varepsilon}{2} \right) - \frac{T^2 d\delta}{2}.
$$

We are now ready to prove our main lower bound.

*Proof.* (of Theorem 4.2) We consider the following construction for the lower bound: the adversary sets $S_{\mathsf{adv}} = (T\varepsilon)^{2/3}$, the sequence of losses will have $E = S_{\mathsf{adv}}^2$ epochs, each of size $B = T/E = T/(T\varepsilon)^{4/3} = \frac{1}{(T\varepsilon)^{1/3}\varepsilon}$. Inside each epoch, the adversary samples $\ell \sim \mathsf{Ber}(1/2)^d$ and plays the same loss function for the whole epoch.

Let $S$ be random variable denoting the number of switches in the algorithm. In this case, we argue that each switch must have a small privacy budget (Lemma C.1), and thus, the price inside each epoch has to be large (Lemma C.2). Let $T_1, \ldots, T_S$ be the rounds where the algorithm resamples ($T_i = T + 1$ for $i > S$) and let $Z_1, Z_2, \ldots, Z_S$ be the resampled models ($Z_i = 1$ for $i > S$). Lemma C.1 implies that

$$
\sum_{i=1}^{T} D_\alpha(Z_i \| Z_i' \mid T_i) \leq \alpha\varepsilon^2.
$$

Now note that inside an epoch $e$, if the algorithm does not switch, then it will suffer loss $B/2$ in that epoch. Otherwise, if it switches, assume without loss of generality there is at most one switch inside each epoch (see Lemma C.2). Let $j_e \in [T]$ denote the index such that $Z_{j_e}$ was sampled in epoch $e$. Note that the algorithm in this epoch has $D_\alpha(Z_{j_e} \| Z_{j_e}' \mid T_{j_e}) = \alpha\varepsilon_e^2$, hence it is $\varepsilon_e^2$-CDP. Standard conversion from concentrated DP to approximate DP (Lemma A.5) implies that it is $(3\varepsilon_e \sqrt{\log(1/\delta)}, \delta)$-DP where $\delta \leq 1/T^3 d$. Hence Lemma C.2 implies the error for this epoch is $B \cdot \left( \frac{1}{2} - \frac{3B\varepsilon_e \sqrt{\log(1/\delta)}}{2} \right) - 1/T$. Letting $E_{switch} \subset [E]$ denote the epochs where there is a switch, we have that the loss of the algorithm is

$$
\begin{aligned}
L(\mathsf{ALG}) &:= \mathbb{E}[\sum_{t=1}^{T} \ell_t(x_t)] \\
&= \mathbb{E}\left[ \sum_{e \notin E_{switch}} \frac{B}{2} \right] \\
&\quad + \mathbb{E}\left[ \sum_{e \in E_{switch}} B \left( \frac{1}{2} - \frac{3B\varepsilon_e \sqrt{\log(1/\delta)}}{2} \right) - 1/T \right] \\
&= \mathbb{E}\left[ (E - S)\frac{B}{2} + S\frac{B}{2} - \sum_{e \in E_{switch}} \frac{3B^2 \varepsilon_e \sqrt{\log(1/\delta)}}{2} - 1 \right]
\end{aligned}
$$

$$= T/2 - 1 - \frac{3B^2\sqrt{\log(1/\delta)}}{2} \mathbb{E}\left[\sum_{e \in E_{switch}} \varepsilon_e\right]$$

$$\geq T/2 - 1 - \frac{3B^2\sqrt{E\log(1/\delta)}\varepsilon}{2},$$

where the last inequality follows since $\sum_{e \in E_{switch}} \varepsilon_e \leq \sqrt{E\sum_{e=1}^{E} \varepsilon_e^2} \leq \sqrt{E}\varepsilon$. Note also that the loss of the best expert is

$$L^\star := \min_{x \in [d]} \sum_{t=1}^{T} \ell_t(x) = T/2 - \sqrt{E}B$$

Overall we get that the regret of the algorithm is

$$L(\mathsf{ALG}) - L^\star \geq \sqrt{E}B - \frac{3B^2\sqrt{E\log(1/\delta)}\varepsilon}{2} - 1$$

$$\geq (T\varepsilon)^{2/3}\frac{T}{(T\varepsilon)^{4/3}} - \frac{3\sqrt{\log(1/\delta)}}{2(T\varepsilon)^{2/3}\varepsilon^2}\sqrt{E}\varepsilon - 1$$

$$= \frac{T^{1/3}}{\varepsilon^{2/3}} - \frac{3\sqrt{\log(1/\delta)E}}{2(T\varepsilon)^{2/3}\varepsilon} - 1$$

$$\overset{(i)}{\geq} \frac{T^{1/3}}{\varepsilon^{2/3}} - \frac{3\sqrt{\log(1/\delta)}}{2\varepsilon} - 1$$

$$\overset{(ii)}{=} \Omega\left(\frac{T^{1/3}}{\varepsilon^{2/3}}\right),$$

where $(i)$ follows since $E \leq (T\varepsilon)^{4/3}$, and $(ii)$ holds since $\frac{3\sqrt{\log(1/\delta)}}{2\varepsilon} \leq \frac{T^{1/3}}{2\varepsilon^{2/3}}$ for $\varepsilon \geq 100\log^{3/2}(dT)/T \geq 27\log^{3/2}(1/\delta)/T$. The claim follows. $\qquad\square$

## C.1 Proof of Lemma C.2

*Proof.* For this lower bound, we assume that the algorithm has full access to $D$ to release $z_1, \ldots, z_T$. First, note that if the algorithms picks $z = z_i$ with probability $1/T$ and releases $(z, \ldots, z)$, then it has the same error since

$$\mathbb{E}[\sum_{t=1}^{T} \ell(z)] = T\,\mathbb{E}[\ell(z)] = T\,\mathbb{E}[\frac{1}{T}\sum_{t=1}^{T}\ell(z_t)] = \mathbb{E}[\sum_{t=1}^{T}\ell(z_t)].$$

Therefore, we assume that the algorithm releases a single $z = \mathsf{ALG}(D)$ that is $(\varepsilon, \delta)$-DP. Denote $D_\ell = (\ell, \ldots, \ell)$. Note that as we sample $\ell \sim \mathsf{Ber}(1/2)^d$, the probability $p := \Pr(\ell = \ell_0) = \Pr(\ell = \ell_1)$ for all $\ell_0, \ell_1 \in \{0,1\}^d$. Letting $\bar{\ell} = 1 - \ell$, we have that

$$\mathbb{E}_{\ell \sim \mathsf{Ber}(1/2)^d}\left[\sum_{t=1}^{T} \ell(\mathsf{ALG}(D_\ell))\right]$$

$$= T \cdot \mathbb{E}_{\ell \sim \mathsf{Ber}(1/2)^d}[\ell(\mathsf{ALG}(D_\ell))]$$

$$= T \cdot \sum_{\ell_0 \in \{0,1\}^d} \Pr_{\ell \sim \mathsf{Ber}(1/2)^d}(\ell = \ell_0) \cdot \mathbb{E}[\ell_0(\mathsf{ALG}(D_{\ell_0}))]$$

$$= \frac{T}{2} \cdot \sum_{\ell_0 \in \{0,1\}^d} p\,\mathbb{E}\left[\ell_0(\mathsf{ALG}(D_{\ell_0})) + \bar{\ell}_0(\mathsf{ALG}(D_{\bar{\ell}_0}))\right]$$

$$\geq \frac{T}{2} \cdot \min_{\ell_0 \in \{0,1\}^d} \mathbb{E}\left[\ell_0(\mathsf{ALG}(D_{\ell_0})) + \bar{\ell}_0(\mathsf{ALG}(D_{\bar{\ell}_0}))\right].$$

Now note that for any $\ell_0$ we have

$$\mathbb{E}\left[\ell_0(\mathsf{ALG}(D_{\ell_0})) + \bar{\ell}_0(\mathsf{ALG}(D_{\bar{\ell}_0}))\right]$$

$$\begin{aligned}
&= \sum_{z \in [d]} \Pr(\mathsf{ALG}(D_{\ell_0}) = z)\ell_0(z) + \Pr(\mathsf{ALG}(D_{\bar{\ell}_0}) = z)\bar{\ell}_0(z) \\
&= \sum_{z \in [d]} \Pr(\mathsf{ALG}(D_{\ell_0}) = z)\ell_0(z) + \Pr(\mathsf{ALG}(D_{\bar{\ell}_0}) = z)(1 - \ell_0(z)) \\
&= 1 + \sum_{z \in [d]} \ell_0(z) \left( \Pr(\mathsf{ALG}(D_{\ell_0}) = z) - \Pr(\mathsf{ALG}(D_{\bar{\ell}_0}) = z) \right) \\
&\geq 1 + \sum_{z \in [d]} \ell_0(z) \left( e^{-T\varepsilon} \Pr(\mathsf{ALG}(D_{\bar{\ell}_0}) = z) - T\delta - \Pr(\mathsf{ALG}(D_{\bar{\ell}_0}) = z) \right) \\
&\geq 1 - Td\delta + \sum_{z \in [d]} \ell_0(z) \Pr(\mathsf{ALG}(D_{\bar{\ell}_0}) = z) \left( e^{-T\varepsilon} - 1 \right) \\
&\geq 1 - Td\delta - \sum_{z \in [d]} \ell_0(z) \Pr(\mathsf{ALG}(D_{\bar{\ell}_0}) = z) T\varepsilon \\
&\geq 1 - Td\delta - T\varepsilon,
\end{aligned}$$

where the first inequality follows since $\mathsf{ALG}$ is $(\varepsilon, \delta)$-DP and group privacy. The claim follows $\quad \square$

