# OpenReview forum: "Private Online Learning via Lazy Algorithms"
_NeurIPS.cc/2024/Conference — NeurIPS 2024 poster_

### Official Review · Reviewer_djv7 · 2024-07-02

**Soundness:** 3
**Presentation:** 4
**Contribution:** 3
**Rating:** 7
**Confidence:** 4

**Summary:**

This paper studies private online prediction from experts (OPE) and online convex optimization (OCO) problems and proposes a general transformation that converts lazy (non-private) algorithms into private algorithms. By applying it to existing lazy algorithms, they obtain improved regret bounds for both problems. A lower bound for lazy algorithms is also provided, suggesting that new techniques other than lazy algorithms are needed if one wants a better regret.

**Strengths:**

1. The proposed method is a general transformation, namely, it applies to any algorithms that satisfy certain properties.
2. The results obtained outperform previous bounds, especially in the high privacy regime ($\varepsilon \ll 1$).
3. A matching lower bound is provided.

**Weaknesses:**

1. The improvement made in this paper may not be very significant. It only improves the dependency on $\varepsilon$, while in many scenarios $\varepsilon$ is treated as a constant.
2. The lower bound may not be essentially matched -- Theorem 4.2 requires the algorithm to be $\varepsilon^2$-CDP instead of $(\varepsilon,\delta)$-DP. It does not mean that every $(\varepsilon,\delta)$-DP algorithm should have that lower bound.

**Questions:**

1. The paper assumes an oblivious adversary. Can the results be extended to the adaptive adversarial setting?
2. It looks like Condition 4.1 is not limited to algorithms with a small number of switches. It only requires that once the algorithm resamples, the resampling distribution depends only on past loss functions and not on internal randomness. Is my understanding correct?

**Limitations:**

Irrelevant.

---

> ### Author Rebuttal · Authors · 2024-08-06
>
> We would like to thank the reviewer for the feedback. We respond to the reviewer’s main comments below; we are glad to clarify further as needed during the discussion period.
>
> Improvement for small $\epsilon$: It is true that in many applications, $\epsilon$ is set to be some constant number. But the online setting is a bit different. Take the DP-OPE, as the non-private regret term $\sqrt{T}$ can dominate the other term, we can get privacy for free even when $\epsilon \ll 1$, and hence the high-privacy regime becomes more important and interesting. This is also crucial in scenarios where many applications of DP-OPE are necessary: in this case, our results show that we can run more such instances of DP-OPE than existing work while satisfying privacy.
>
> Lower bound for CDP: Lower bounds in the oblivious setting turn out to be elusive, and prior work has only resulted in the trivial lower bound of $1/\epsilon$. Therefore, we restricted ourselves to the family of low-switching algorithms. The choice of CDP is made so that it is easier to prove the tight composition of the privacy budget. We can prove similar (tight) lower bounds for pure DP as it has a simple tight composition (our upper bounds are tight for pure DP).
>
>
> Question 1: The privacy guarantee may be extended to adaptive adversaries, but the utility guarantee may be broken. As the algorithm is low-switching with the same model choice over a batch, the adaptive adversarial may design some loss functions with very large regret based on this. We note that our main goal in this paper is to study oblivious adversaries, especially given the recent work [AFKT23], which studies adaptive adversaries and gives tight lower bounds for certain privacy regimes.
>
> Question 2: Yes, you are right. Our lower bounds should hold for this family of algorithms as well.

---

> > ### Comment · Reviewer_djv7 · 2024-08-09
> >
> > Thank you for your addressing my questions. I decide to keep my positive score.

---

### Official Review · Reviewer_MEJp · 2024-07-11

**Soundness:** 3
**Presentation:** 3
**Contribution:** 3
**Rating:** 6
**Confidence:** 4

**Summary:**

This paper investigates private online learning, focusing on online prediction from experts (OPE) and online convex optimization (OCO). The authors propose a new transformation that translates lazy, low-switching online learning algorithms into private algorithms. Based on this transformation, their resulting algorithms attain regret bounds that significantly improve over prior art in the high privacy regime. Furthermore, they complement their results with a lower bound for DP-OPE, showing that these rates are optimal for a natural family of low-switching private algorithms.

**Strengths:**

* The improvements in results for DP-OPE and DP-OCO are highly valuable.
* The research motivation of this work is clear, and the proposed transformation (L2P) is a very solid contribution to DP online learning.

**Weaknesses:**

* The transformation in this work is based on low-switching online algorithms, and is only applicable to oblivious adversaries. In contrast, previous DP-OCO algorithms [ST13, AS17, KMS+21]  can adapt to non-oblivious setting.
* The theoretical innovation in this paper is limited, as it largely follows previous work [AFKT23b].

**Questions:**

* In footnote 2, the authors state that their algorithms will satisfy a stronger notion of differential privacy against adaptive adversaries. However, I believe that for non-oblivious adversaries, the proposed DP online algorithms in this work cannot provide regret guarantees. Is this correct? If so, I would like to understand why the low-switching private algorithms cannot handle non-oblivious adversaries, whereas previous work using binary tree techniques [ST13, AS17, KMS+21]  can.
* From Theorem 3.2, it is evident that $\epsilon$ of L2P increases over time, meaning that its privacy weakens progressively. Could this be a concern for the proposed DP algorithm?
* The authors established a lower bound for low-switching private online algorithms based on **CDP** analysis, whereas the upper bound provided in this paper is based on **DP** analysis, i.e, Theorem 3.2 and 3.9. Could you clarify the implications of this difference?

**Limitations:**

See above

---

> ### Author Rebuttal · Authors · 2024-08-06
>
> We would like to thank the reviewer for the feedback. Please see our responses below; we are glad to clarify further as needed during the discussion period.
>
> Adaptive adversary: we note that our main focus in this work is oblivious adversaries since the recent work of [AFKT23] studies adaptive adversaries and shows tight lower bounds for certain privacy regimes. Given the separation in rates between adaptive and oblivious adversaries, the goal of our work is to study and understand the fundamental limits of oblivious adversaries.
>
> Innovation: Some works in this line implicitly show the connection between lazy algorithms and private algorithms, but we are the first to formalize this connection explicitly via our L2P framework that can convert general lazy algorithms into private ones. The framework does not follow previous works like [AFKT23b] directly and requires new techniques and analysis, such as the new correlated sampling strategy through a parallel sequence of models, or the new regret guarantees that measure the effect of batching on lazy online algorithms.
>
>
> We also answer the reviewer’s questions below.
>
> Q1: This is correct. Our regret bound may be invalid with an adaptive adversary, and there are some challenges in extending it further. For example, as our algorithm consistently produces output over one batch, the adversary can know our next $B$ predictions and might choose some bad loss functions. The privacy guarantee, however, holds against adaptive adversaries. This is similar to several results in DP ML, where we get privacy for any input sequence, while utility holds under some distributional assumptions. In other words, if our assumptions are invalid, we may get worse utility but will not lose privacy.
>
> Q2: We will lose the privacy budget each time we make a switch, and it is unavoidable to weaken the privacy guarantee. However, we set the privacy budget for each time step in a way that guarantees that the final privacy parameter guaranteed by the algorithm is satisfactory. This requires that the number of rounds $T$ is bounded, as is the case in most private optimization procedures: for example, in DP-SGD with full batch size, the privacy weakens over iterations as well, and therefore we need to bound $T$.
>
> Q3: Lower bound for CDP:  lower bounds in the oblivious setting turn out to be elusive and prior work had only resulted in the trivial lower bound of $1/\varepsilon$. Therefore, we restricted to the family of low-switching algorithms. The choice of CDP is made so that it is easier to prove the tight composition of the privacy budget. We can prove similar (tight) lower bounds for pure DP as it has a simple tight composition (our upper bounds are tight for pure DP).

---

> > ### Comment · Reviewer_MEJp · 2024-08-08
> >
> > Thanks for your response, which has addressed my concerns. I decide to keep my positive score.

---

### Official Review · Reviewer_rLS6 · 2024-07-12

**Soundness:** 3
**Presentation:** 3
**Contribution:** 3
**Rating:** 6
**Confidence:** 3

**Summary:**

The paper studies the differentially private (DP) variants of the classical online prediction form experts problem (OPE) and online convex optimization (OCO) problem. The main contribution is a (black-box) approach to transform lazy (i.e. slow-varying) online learning algorithms into private algorithms. The paper shows that the transformation only incurs a small additional regret as compared to the regret for the classical OPE/OCO regret. In particular, for the  DP-OPE and DP-OCO, the proposed algorithms improve the O(1/eps) regret in previous works to O(1/eps^{2/3}). In addition, a matching regret lower bound is provided for the family of low-switching private algorithms for the DP-OPE problem.

**Strengths:**

The paper is well-written with all claims supported by proofs. Compared to previous works, the L2P algorithm proposed uses a new correlated sampling strategy to avoid accumulation of privacy cost, and the paper improves the regret of lazy algorithms due to batching. In addition, the paper provides a lower bound for the DP-OPE problem, showing that limited switching is not enough to obtain faster rate, suggesting the need for other techniques.

**Weaknesses:**

The lower bound is only valid for the family of slow-varying algorithms, thus the algorithms proposed might not be optimal.

**Questions:**

The paper is relatively dense, with many definitions related to differential privacy (section 2), which might be unfamiliar for people who have little knowledge about differential privacy before. I’m wondering if all those definitions are necessary for the main part of the paper (maybe some can be moved to the appendix)?

**Limitations:**

Yes the paper has discussed the limitations and provides improvement in the appendix.

---

> ### Author Rebuttal · Authors · 2024-08-06
>
> Thanks for your review and feedback. Below, we address the main comments; we are glad to clarify further as needed during the discussion period.
>
> 1. Lower bounds: we acknowledge that the conditional lower bound is not ideal, but it is important to note that this is the first non-trivial (even conditional) lower bound for oblivious adversaries. Several papers have studied DP-OPE and DP-OCO, and so far, none has come up with a better lower bound than the trivial $1/\varepsilon$ lower bound (though for adaptive adversaries, there are some lower bounds). We, therefore, believe that our conditional lower bound (while admittedly not entirely satisfactory) is a step in the right direction towards proving general non-trivial lower bounds for oblivious adversaries. Finally, we note that our lower bound proves that our rate is the best possible using existing techniques that have been pursued by the recent line of work based on limited switching.
>
> 2. Organization: we will reorganize the paper, add more discussions, and defer less important definitions and details to the appendix, in order to make the paper more readable for people less familiar with differential privacy.

---

> > ### Comment · Reviewer_rLS6 · 2024-08-08
> >
> > Thank the authors for the additional comments on the lower bound! I will keep my score!

---

### Official Review · Reviewer_FJx4 · 2024-07-22

**Soundness:** 3
**Presentation:** 3
**Contribution:** 3
**Rating:** 6
**Confidence:** 3

**Summary:**

This paper proposes a new mechanism that converts lazy online learning algorithms into private algorithms. Unlike previous private online algorithms that use individualized privatized methods, the paper's new mechanism is a black box private algorithm that can be applied to many popular non-private methods such as the lazy shrinking dartboard algorithm or the regularized multiplicative weights algorithms.

**Strengths:**

- The problem is well-motivated. Even though previous private algorithms have good theoretical guarantees, their privatized mechanism is usually tailor-made to solve a specific problem, which limits their applicability. The new black box algorithm makes converting a non-private to a private one simple and seamless.

-  The private term has improved dependence in the $\epsilon$ term which could be valuable in the high privacy regime (where $\epsilon \ll 1$).

- The authors also provide lower bounds that match their upper bound.

- The paper is well-written overall.

**Weaknesses:**

- I'm not sure if I agree with the claim that the new bound is an improvement over the previous bound. Obviously, the high privacy regime is very important, and as the authors have mentioned, this would allow for the composition of more instances. However, as far as I know, a lot of practical application of DP algorithm uses $\epsilon \approx O(1)$ so the new bound could be worse than previous results.

- It would be nice if there are some proof of concept experiments to compare the new mechanism with the algorithms in previous works.

**Questions:**

- Can the author provide some intuition on why we would want to use concentrated DP to prove the lower bound? How does concentrated DP make it easier for us to construct the lower bound?

- What are some non-lazy algorithms that satisfy Assumption 3.1? What would the regret bound look like for these non-lazy algorithms.

- I'm a bit confused about the setting of $\epsilon$ in Theorem 3.2. To get the optimal regret bound, we need to set $\epsilon$ using the provided formula which I think is roughly $O(1)$. Doesn't that defeat the point of the proposed framework where we only see improvement when $\epsilon <<1$.

---

> ### Author Rebuttal · Authors · 2024-08-06
>
> Thanks for your time in reviewing and feedback. Please see our responses below; we are glad to clarify further as needed during the discussion period.
> - Improvement for small $\varepsilon$: First, by modifying the parameter settings, we can recover the previous best results when $\epsilon \ge \Omega(1)$; second, as the non-private regret $\sqrt(T)$ can dominate the private cost in regret when $\epsilon$ is large, the high privacy regime is interesting and non-trivial. In practice, people are usually forced to use constant epsilon as smaller values of epsilon degrade the performance significantly. However, in our setting, we can get privacy for free (when the non-private regret $\sqrt{T}$ dominates) even if $\epsilon \ll 1$. Therefore finding the smallest $\varepsilon$ (or best privacy) possible that allows the non-private regret $\sqrt{T}$ is an interesting open question, and our work provides progress towards resolving it.
>
> - Q1, Intuition for concentrated DP: Lower bounds in the oblivious setting turn out to be elusive, and prior work has only resulted in the trivial lower bound of $1/\varepsilon$. Therefore, we restricted ourselves to the family of low-switching algorithms. The choice of CDP is made so that it is easier to prove the tight composition of the privacy budget. We can prove similar (tight) lower bounds for pure DP as it has a simple tight composition (our upper bounds are tight for pure DP).
>
> - Q2, non-lazy algorithms: The classic Multiplicative Weights algorithm (that draws a fresh sample at each iteration) may be non-lazy, but still satisfies Assumption 3.1. As long as it satisfies Assumption 3.1, we can prove the regret bound as claimed.
>
> - Q3, setting of $\varepsilon$ in Theorem 3.2: The theorem provides a way to calculate $\varepsilon$ based on the parameters of the algorithms such as $\eta$ and $p$. As a result, we can tolerate much smaller values of $\varepsilon$: indeed, in Theorem 3.9 and Theorem 3.11, we instantiate Theorem 3.2 with several hyperparameters that result in small values of $\varepsilon$. Hope this clarifies your concern.

---

### Decision · Program_Chairs · 2024-09-25

**Decision:**

Accept (poster)

**Comment:**

This manuscript studies the problem of online prediction from experts (OPE) and online convex optimization (OCO) under the differential privacy constraints. The main contribution of this work is that, instead of applying individualized privatized methods, it proposes a new mechanism that converts lazy online learning algorithms into private algorithms in a black-box manner. The significance is two-fold: first, this new mechanism improves the best-known regret bounds for private OPE and OCO problems; second, this new mechanism can be applied to many other non-private methods and thus has a great potential to generalize. This idea is greatly appreciated by all reviewers, and so leads to my decision of acceptance.